# Fabrication and Oxidation Resistance of a Novel MoSi_2_-ZrB_2_-Based Coating on Mo-Based Alloy

**DOI:** 10.3390/ma16165634

**Published:** 2023-08-15

**Authors:** Yafang Zhang, Xiaojun Zhou, Huichao Cheng, Zhanji Geng, Wei Li

**Affiliations:** 1State Key Laboratory of Powder Metallurgy, Central South University, Changsha 410083, China; 203101037@csu.edu.cn (Y.Z.); zxj745301@126.com (X.Z.); cheng26@csu.edu.cn (H.C.); gengzhanj@163.com (Z.G.); 2School of Materials Science and Engineering, Central South University, Changsha 410083, China

**Keywords:** MoSi_2_-ZrB_2_ composite coating, multi-layer structure, oxidation resistance, diffusion barrier

## Abstract

To enhance the oxidation resistance of Mo-based TZM alloy (Mo-0.5Ti-0.1Zr-0.02C, wt%), a novel MoSi_2_-ZrB_2_ composite coating was applied on the TZM substrate by a two-step process comprising the in situ reaction of Mo, Zr, and B_4_C to form a ZrB_2_-MoB pre-layer followed by pack siliconizing. The as-packed coating was composed of a multi-layer structure, consisting of a MoB diffusion layer, an MoSi_2_-ZrB_2_ inner layer, and an outer layer of mixture of MoSi_2_ and Al_2_O_3_. The composite coating could provide excellent oxidation-resistant protection for the TZM alloy at 1600 °C. The oxidation kinetic curve of the composite coating followed the parabolic rule, and the weight gain of the coated sample after 20 h of oxidation at 1600 °C was only 5.24 mg/cm^2^. During oxidation, a dense and continuous SiO_2_-baed oxide scale embedded with ZrO_2_ and ZrSiO_4_ particles showing high thermal stability and low oxygen permeability could be formed on the surface of the coating by oxidation of MoSi_2_ and ZrB_2_, which could hinder the inward diffusion of oxygen at high temperatures. Concurrently, the MoB inner diffusion layer played an important role in hindering the diffusion of Si inward with regard to the TZM alloy and could retard the degradation of MoSi_2_, which could also improve the long life of the coating.

## 1. Introduction

Molybdenum (Mo) alloys have been considered as a promising candidate for high-temperature structural materials owing to their high melting points, outstanding mechanical properties at elevated temperatures, low coefficient of thermal expansion (CTE), high thermal conductivity, and good thermal shock resistance [1,2,3]. However, poor high-temperature oxidation resistance of molybdenum alloys limits their applications in an oxidizing environment at high temperatures [3,4,5]. It has been recognized that oxidation-resistant coatings could offer an effective approach to improve the oxidation resistance of Mo-based alloy at elevated temperatures. An effective high-temperature oxidation-resistant coating must be capable of forming a protective oxygen barrier at high temperatures. Dense silica scale provides excellent resistance to oxygen diffusion or permeation at high temperatures. Thus, great attention has been currently paid to materials based on molybdenum disilicide (MoSi_2_) for use as a protective coating against high-temperature oxidation on Mo-based alloys, due to its high melting point (2030 °C), moderate density of 6.24 g/cm^3^, similar CTE (8.1 × 10^−6^/°C) to the Mo substrate (6.7 × 10^−6^/°C), and the regenerating of a protective SiO_2_ scale on the coating surface [6,7,8]. However, MoSi_2_ performs limited protective performance in higher temperature applications (e.g., up to 1600 °C), mainly related to two factors: one is the decrease in the thermal stability of SiO_2_ at higher temperatures, and the other is the thermal–mechanical limitations of MoSi_2_ induced by the rapid interdiffusion between the coating and the substrate and the thermal stresses generated by the CTE mismatch at the coating–substrate interface [9].

To improve the thermal stability of the SiO_2_ oxide scale on the MoSi_2_ coating, one of the research hotspots is to develop MoSi_2_-based composite coatings doped with alloying elements (e.g., B [10,11], Al [12,13], Ti [14], Zr [15], and Hf [13]) or/and oxides (e.g., Al_2_O_3_ [16,17,18,19,20], ZrO_2_ [18,21], HfO_2_ [18,22]), carbides (e.g., ZrC [23]), and borides (e.g., ZrB_2_ [24,25]), which have been confirmed to be effective in improving the performance and longevity of the MoSi_2_ coating at elevated temperatures. The B-doped MoSi_2_ coating has been proven to perform excellent oxidation resistance benefiting from the formation of a protective borosilicate scale on the coating [10,11]. Zr and Zr-based compounds (e.g., ZrC, ZrB_2_) have been recognized to be a suitable additive in improving the high-temperature performance of the MoSi_2_ coating due to the formation of a composite oxide scale based on SiO_2_, ZrO_2_, and ZrSiO_4_ with enhanced thermal stability compared to a pure SiO_2_ scale [23,24,25].

Zirconium diboride (ZrB_2_) is typical ultra-high-temperature ceramic and has been widely studied for high-temperature structural applications owing to its excellent and unique combination of high melting points exceeding 3000 °C, good thermo-chemical property, high thermal conductivity, and ablation resistance [26,27,28]. As with other non-oxide ceramics, ZrB_2_ will be oxidized when exposed to air at elevated temperatures and generate oxidation products of ZrO_2_ and B_2_O_3_. However, since B_2_O_3_ would volatilize at temperatures above 1200 °C, a pure ZrO_2_ scale would be formed by oxidation of ZrB_2_ at high temperatures up to 1200 °C [29]. The ZrO_2_ scale has a porous structure and could not act as a diffusion barrier of oxygen; thus, the single ZrB_2_ coating cannot be directly applied to oxidation and ablation protection of high-temperature structural materials. Adding silicon compounds, such as MoSi_2_ and SiC, is an effectively way to solve this problem. There are a large number of reports concerning the fabrication of ZrB_2_-based structural ceramics and functional coatings, such as ZrB_2_-SiC [30], ZrB_2_-MoSi_2_ [31,32]. However, there are hardly any studies on the ZrB_2_-doped MoSi_2_ based coating on Mo-based alloy.

The lifetime of the MoSi_2_ coating was limited by the Si depletion induced by the interdiffusion between the MoSi_2_ coating and the refractory metals substrate. Applying a diffusion barrier between the MoSi_2_-based coating and the refractory metals substrate is another way to improve the stability and serving life of the coating. Many kinds of materials, including oxides (Al_2_O_3_ diffusion barrier for the Mo-Si-B/Nb-Si system [33], Y_2_O_3_ diffusion barrier for the MoSi_2_/Nb-Si system [34]), carbides (Mo_2_C diffusion barrier for the MoSi_2_/SiC-Mo_2_C/Mo system [35], SiC-glass diffusion barrier for the MoSi_2_/Nb-Si system [36]), borides (MoB diffusion barrier for the MoSi_2_/Mo system [8,37], Nb_3_B_2_-NbB_2_ diffusion barrier for the Si-Mo-W-ZrB_2_-Y_2_O_3_/Nb system [38]) and silicides (TaSi_2_ diffusion barrier for the MoSi_2_/Nb system [39], WSi_2_ diffusion barrier for the MoSi_2_/Nb-Ti-Si system [40]) have been used as diffusion barriers between the coatings and the substrates, thereby effectively lengthening the lifetime of these coatings. Reference [25] demonstrated that the Mo_2_C barrier layer plays a key role in slowing down the diffusion of C and Si toward the inner Mo substrate at high temperatures, which led to less cracks in the surface and a thinner Mo_5_Si_3_ layer at high temperatures in air and thus brought about better oxidation resistance. References [8,34] reported that the MoSi_2_/MoB coating (MoB as a diffusion barrier) possesses a longer service life compared to the single MoSi_2_ coating on the Mo substrates, owing to the diffusion barrier effect of the MoB layer on the Si element diffusion into the Mo substrate.

In this work, a composite coating, composed of the MoSi_2_-ZrB_2_ main layer and the MoB diffusion layer between the coating and substrate, was fabricated on the TZM alloy substrate by a two-step method, including a slurry sintering step of Mo, Zr, and B_4_C, followed by a halide-activated pack cementation process. Isothermal oxidation behavior of the composite coating was evaluated at 1600 °C in air. Microstructure and phase composition of the as-prepared and oxidized composite coating were characterized, and the antioxidant mechanism of the coating at high temperatures was also discussed. In contrast, the structure evolution and oxidation resistance of a single MoSi_2_ prepared on the TZM alloy through the same two-step method were studied as well.

## 2. Materials and Methods

### 2.1. Sample Preparation

TZM Mo-based alloy (Mo-0.5Ti-0.1Zr-0.02C, wt%) was used as substrates in the study. Long strip specimens (90 mm × 8 mm × 2 mm) were cut from the TZM alloy plate. All samples were hand-polished by SiC grit papers up to 1000 mesh and cleaned in an ultrasonic ethanol bath and then dried at 80 °C for 2 h in vacuum.

The composite coating was prepared on the TZM alloy substrates through a two-step process:(1)Firstly, two kinds of slurry were prepared. Mo powders (purity > 99.5%, 1~2 μm) and ethyl acetate (as solvent, AR) were putted into a ball milling tank, and the weight/volume ratio of Mo powders to solvent (g:mL) was 1:0.8. A small amount of nitrocellulose (as organic binder, AR) was added to the ball milling tank. The mixture was attrition milled for 6 h to obtain a Mo slurry. Concurrently, a Zr-B_4_C slurry was prepared by mixing of pure Zr powders (purity > 99%, 2–3 μm), B_4_C powders (purity > 99.5%, 2–3 μm), and ethyl acetate. The molar ratio of Zr:B_4_C was 1:1, while the weight/volume ratio of powders to solvent (g:mL) was also 1:0.8. In the same way, a small amount of nitrocellulose (AR) was added to the mixture. Then, the mixture was also attrition milled for 6 h.

Secondly, the Mo slurry was sprayed evenly on the surface of the Mo-based alloy substrate. The as-sprayed samples were dried using an infrared baking lamp and a Mo pre-layer was obtained on the substrate. Subsequently, the Zr-B_4_C slurry was sprayed evenly on the surface of the Mo pre-layer and then dried using infrared baking lamp as well.

Finally, the samples were taken into a vacuum furnace and sintered at 1700 °C for 2 h in a vacuum (<1 Pa). Thus, a Mo/Zr-B_4_C ceramic pre-layer was fabricated on the Mo-based alloy substrate. To simplify the expression, the as-ceramic pre-layer will be referred to as the “Mo/Zr-B_4_C ceramic pre-layer” in this paper.

(2)Subsequently, a conventional pack cementation process was carried out. The pack mixture consisted of Si powders (pack cementation element, 99.9% purity, 1~3 μm), Al_2_O_3_ powders (inert powder, 98% purity, ~75 μm), and NaF powders (activator, 99.9% purity, 1~2 μm) with a mass ratio of 67:30:3. The pack powders were mixed up by tumbling in a ball mill for 12 h and dried at 110 °C for 6 h. The samples containing ceramic pre-layer on surface were embedded in the pack mixture and then heated to 1150 °C and held for 10 h in an argon-protected tube furnace. Finally, the sintered coating samples were furnace-cooled down to room temperature.

For comparison, a pure MoSi_2_ coating on the TZM alloy was prepared through the same two-step method: (1) slurry sintering to obtain a Mo pre-layer on the TZM alloy substrate and (2) siliconizing with Si powders to generate a MoSi_2_ coating on the substrate.

### 2.2. Oxidation Test and Characterization

Isothermal oxidation tests were performed at 1600 °C in static air using an electric furnace to investigate the oxidation resistance of both kind of coatings. Weight gains of both coatings after oxidation for different time were measured using an analytical balance with an accuracy of 10^−4^ g to study the oxidation characteristics of the coatings.

The coated specimens before and after oxidation were wire cutting and acetone cleaned. Phase composition of the surface of the as-prepared and oxidized coatings was analyzed by X-ray diffraction (XRD, D/Max 2500, Cu-Ka radiation, Tokyo, Japan). The surface of the coatings before and after oxidation was obtained using scanning electron microscopy (SEM, TESCAN MIRA3, Brno, Czech Republic) coupled with an energy dispersive spectrometer (EDS) device. Since the surface of the oxidized coating was not conductive after high-temperature oxidation, it needed to be sprayed with gold before observation. The sample section was embedding by bakelite at 175 °C with the pressure of a 175 bar. After sanding (using 200 mesh, 400 mesh, 600 mesh and 800 mesh sandpaper) and polishing, the cross-sectional microstructure and elemental distribution of the as-prepared and oxidized coatings were characterized by SEM and an electron-probe micro-analyzer (EPMA, JEOL JXA-8230, Musashino, Japan) equipped with wave dispersive X-ray spectroscopy (WDS).

## 3. Results and Discussion

### 3.1. Microstructure and Phase Composition of the Coating

#### 3.1.1. Microstructure and Phase Composition of Ceramic Pre-Layer

Figure 1 displays the XRD pattern of the surface of the sintered ceramic pre-layer. According to the XRD pattern, the as-fabricated ceramic pre-layer was composed of ZrB_2_ and MoB. Carbide (e.g., ZrC, Mo_2_C, MoC) was not obtained in the sintered samples. The diffraction peaks belonging to ZrB_2_ were shifted toward larger angles, which are potentially attributed to dissolution of C in the ZrB_2_ lattice. No initial component such as Zr, Mo, or B_4_C was detected in the sintered samples, indicating sufficient reactions between Zr, Mo, and B_4_C. The presence of ZrO_2_ in the sintered pre-layer could be attributed to the oxide impurities on the surface of the strating powders of Zr and B_4_C. Meanwhile, Zr and B_2_O_3_ would react and generate ZrO_2_ from 5Zr + 2B_2_O_3_ = 3ZrO_2_ + 2ZrB_2_ [41,42,43]. However, in our previous work, ZrO_2_ proved to be beneficial to the high temperature oxidation resistance of silicide coating [44].

In order to further investigate the phase morphology and elemental distribution in the ceramic pre-layer, the typical microstructure and element mapping of the surface morphology of the ceramic pre-layer are given in Figure 2a–f. As shown in Figure 2a, the coating had a relatively loose surface and showed an island-like morphology. There were numerous pores around the island-like clusters. However, no visual crack was present on the surface of the sintered ceramic pre-layer. Figure 2b shows an enlarged view of area A in Figure 2a. It is clear that those island-like clusters became intermingled with each other, which could offer a good interlocking of the coating against peeling off. Figure 2c–f show the element mapping of the surface of the sintered ceramic pre-layer in Figure 2b. It can be seen that the pre-layer mainly contained Zr, Mo, B, and C elements. Given the XRD pattern in Figure 1 and the element mapping in Figure 2c–f, we concluded that the island-like clusters were composed of ZrB_2,_ in which some of the MoB phases were distributed uniformly.

Figure 3a shows the cross-sectional microstructure of the ceramic pre-layer. From a cross-sectional perspective, the ceramic pre-layer represented a loose but interconnected morphology, which had 100~110 μm of thickness. No visible crack was found in the cross section either. Figure 3b–d show the element mapping of the cross section of the pre-layer. The ceramic pre-layer regions in Figure 3a match well with Zr-rich regions in Figure 3b. There was no obvious diffusion layer between the ceramic pre-layer and the substrate. It can be seen from Figure 3c,d that the MoB phases were distributed uniformly throughout the depth, concurrently, although B was likely to diffuse into the Mo-based alloy substrate besides in the pre-layer.

For the Zr-B_4_C system, when sintering at temperatures above 900 °C, B_4_C could react with metals (Zr and Mo) to generate corresponding borides and carbides, and the main reaction products usually include ZrB_2_ and ZrC [45,46]. The possible reactions for the Zr-B_4_C system are listed as Reactions (1)–(4). For a certain reaction, it could proceed positively only when the Gibbs free energy ΔG < 0. The standard Gibbs free energy at a certain temperature T could be calculated, which is given in Figure 4. It is clear that Reactions (1)–(4) are all thermodynamically favorable in the calculated temperature range. The standard Gibbs free energy of Reaction (1) was more negative than that of Reaction (2). On basis of the principle of minimum free enthalpy, the Reaction (1) was more likely to occur. During the solid reaction process of the Zr-B_4_C system, Zr would firstly react with B_4_C to generate ZrB_2_ and amorphous carbon (C), and then amorphous carbon reacted with residual Zr to form ZrC by Reaction (4) [35]. Rehman S.S. et al. [36] also proved the formation of ZrB_2_ and element C during spark plasma sintering of B_4_C and ZrH_2_ mixture powders. However, in the present work, the sintered ceramic pre-layer was mainly composed of ZrB_2_ and MoB, and no ZrC, MoC, or Mo_2_C phase was detected by XRD according to Figure 2. It is supposed that the formation of ZrC from the in situ reaction of Zr and B_4_C is related to the reaction temperature and the ratio of Zr/B_4_C. For one thing, the sintering temperature in this work was 1700 °C. Wu W. et al. [39] reported the reaction behavior of ZrH_2_ and B_4_C at a temperatures range from 900 to 1600 °C. They revealed that ZrH_2_ decomposed to Zr and H_2_(g) at 900 °C. At a temperature range from 900 °C to 1000 °C, Zr would react with B_4_C to form ZrC and ZrB_2_, and when the temperature increased to 1200 °C and 1400 °C, the XRD peak intensity of ZrC decreased, whereas the ZrB_2_ peak intensity increased. Upon a heat treatment at 1600 °C, the ZrC phase completely disappeared and only ZrB_2_ was detected. The occurrence of chemical reactions between ZrB_2_ and ZrC at temperatures above 1500 °C has also been reported, indicating the carbon in ZrC would be substituted by boron with a further increase in temperature [47]. Moreover, Chen H.A. et al. [42] reported the formation mechanism of ZrB_2_-ZrC-B_4_C ceramics from the in situ reaction of Zr-B_4_C; according to their analysis, since the formation of ZrB_2_ was a priority and rapid, and the free energy of the ZrC formation was relatively higher than that of ZrB_2_, the formation of ZrC depended on the content of Zr. No ZrC was obtained below 50 vol.% Zr content. In this work, the molar ratio Zr/B_4_C was 1:1 (~38 vol.% Zr), and due to the lack of Zr, amorphous carbon could not react with Zr to generate ZrC and existed in the pre-layer in a solid solution. The atomic radius of the C atom (0.77 Å) was smaller than that of the B atom (0.97 Å), and the presence of the interstitial C atom in the ZrB_2_ lattice caused a change in the lattice constants, which led to the ZrB_2_ diffraction peaks shifting toward larger angles in the XRD pattern, as shown in Figure 1.

For the Mo/Zr-B_4_C system, besides Reactions (1)–(4), the potential chemical reactions are listed as (5)–(8). The standard Gibbs free energy for Reactions (5)–(8) at a certain temperature T has also been calculated and shown in Figure 4. Similarly, the standard Gibbs free energy of Reaction (5) was more negative than that of Reaction (6), implying the priority formation of MoB rather than Mo_2_C, which is consistent with the XRD result in Figure 1.
Zr + B_4_C = ZrB_2_ + C (1)
Zr + B_4_C = ZrC + B (2)
Zr + B= ZrB_2_
(3)
Zr + C = ZrC(4)
Mo + B_4_C = MoB + C(5)
Mo + B_4_C = Mo_2_C + B(6)
Mo + B = MoB (7)
Mo + C = Mo_2_C(8)

#### 3.1.2. Microstructure and Phase Composition of the Coating

The ceramic pre-layer samples were pack siliconized further to obtain the MoSi_2_-based composite coating. Figure 5a displays the XRD pattern of the surface of the composite coating. The XRD pattern shows the peaks of MoSi_2_ (JCPDS No. 80-0544) and ZrB_2_ (JCPDS No. 89-3930). The diffraction peaks belonging to MoSi_2_ and ZrB_2_ were both shifted toward the larger angles. According to the above analysis, the X-ray diffraction peaks of the ZrB_2_ phase with the solution of C shifted toward the larger angles. It is supposed that C or/and B dissolved in the MoSi_2_ phase. Since the atomic radius of the B and C atoms are smaller than that of Si atom (1.18 Å), the peak of MoSi_2_ also shifted toward lager angles. The ZrO_2_ phase in the sintered ceramic pre-layer was not detected in the as-packed coating probably because of low content. It is noted that the MoB phase in the sintered ceramic pre-layer was also absent in the as-packed coating due to the chemical reaction with Si to form MoSi_2_, which will be discussed below. It can be seen from Figure 5b that only MoSi_2_ (JCPDS No. #80-0544) was detected for the MoSi_2_ coating.

Figure 6a–c show the surface morphology of the as-prepared MoSi_2_-ZrB_2_ coating. Element mapping of the surface of the coating in Figure 6b is displayed in Figure 6d–i. As shown in Figure 6a, the surface of the coating was more compact compared to the ceramic pre-layer in Figure 2a. However, the coating surface was relatively loose and rough. Island-shape clusters were dispersed uniformly on the entire coating surface. An enlarged view of area B in Figure 6a was presented in Figure 6b. As shown in the picture, the average diameter of these clusters was approximately 20 μm, and they were closely adhered to the other clusters around them. Some flaky particles were inlaying in the clusters, and were supposed to be Al_2_O_3_ according to element mapping in Figure 6h,i, which was originated from the pack mixture during pack cementation. Al_2_O_3_ has been proved to be beneficial to the high-temperature oxidation resistance of the MoSi_2_ coating in our previous work [18,19]. Nevertheless, Al_2_O_3_ was not detected by XRD in Figure 5 due to low content. An enlarged view of area C in Figure 6b is given in Figure 6c. Combined with Figure 6d–g, it is clear that a number of the ZrB_2_ particles with size of about 1~3 μm were distributed on the surface of the MoSi_2_ clusters, which could be confirmed by XRD in Figure 6 and element mapping in Figure 6f,g.

Figure 7a show the cross-sectional microstructure of the as-prepared coating. The coating displayed a multi-layer structure, consisting of an outer layer (layer I in Figure 7a), an inner layer (layer II in Figure 7a) and an interdiffusion layer (layer III in Figure 7a) from outside to inside, with thicknesses of about 20~24 μm, 57~67 μm and 5~7 μm, respectively. WDS analysis revealed that the composition of Spot 1 in Figure 7a is 31.45Mo-66.09Si-0.83Zr-0.53B-0.67Al-0.43O (at.%), and it was identified to be MoSi_2_. The composition of Spot 2 was similar to that of Spot 1, which was composed of MoSi_2_ as well. Black phases (Spot 3) were observed inlaid in the outer layer and were determined to be Al_2_O_3_ by WDS analysis. The diffusion layer (layer III) between the coating and substrate mainly contained the Mo and B elements and was inferred to be MoB since the ratio of Mo and B was ~1:1 by WDS analysis. Figure 7b–f show the element mapping of the cross section of the as-prepared composite coating, which further confirms the phase distribution and element composition of the coating. It can be seen that the Si elements were mainly distributed in inner layer (layer II) and outer layer (layer I), while the diffusion layer (layer III) was rich in B and poor in Si. The Zr elements were mainly distributed in the inner layer. Combined with XRD pattern and EPMA analysis, it is concluded that the outer layer (layer I) consisted of MoSi_2_ and Al_2_O_3_, and the inner layer (layer II) comprised a mixture of MoSi_2_ and ZrB_2_, and the diffusion layer (layer III) was MoB.

Microcracks were found at the interface between the MoB diffusion layer and MoSi_2_ layer, which was expected to form during the cooling down from the sintering temperature to room temperature, mainly due to the difference of the coefficient of thermal expansion (CTE) between and MoB and MoSi_2_. Thermal stress resulting from the CTE mismatch between the interface could cause cracks in the coating if they reached or exceeded a critical value of the strength of MoSi_2_. The thermally induced stress on the cooling can be approximated calculated by the equation as follow [48].
(9)$$\sigma_{therm}=\frac{E_{C}∆T(\alpha_{S}-\alpha_{C})}{1-\upsilon_{C}}$$
where $\sigma_{therm}$ is the thermal induced stress, E is the Young’s modulus, ΔT is the temperature difference, α is the CTE, υ is the Poisson’s ratio, and the subscripts C and S represent the coating and substrate. The thermal stresses developed in the MoSi_2_ coating are 1041 MPa and 441 MPa for Mo and MoB, respectively, which definitely exceed the tensile strength of MoSi_2_ (275 MPa) [18,48,49].

During siliconization, Si would react with NaF to generate active Si atoms via catalysis of NaF, which was favorable for the formation of the MoSi_2_-based coating due to the lower activation energy of the Si diffusion [50]. The possible reactions during siliconization are listed in Reactions (10)–(14):2Si + MoB = MoSi_2_ + B(10)
3Si + 5Mo = Mo_5_Si_3_(11)
2Si + Mo = MoSi_2_(12)
7Si + Mo_5_Si_3_ = 5MoSi_2_(13)
B + Mo = MoB(14)

Figure 8 shows the values of the Gibbs free energy for the Reactions (10)–(14). All the Reactions are thermodynamically favorable in the calculated temperature range. During the siliconization process, active Si started to deposit on the surface of the samples and then diffused into the ceramic pre-layer and reacted with the ceramic pre-layer to form silicides by Reaction (10). Because the structure of the ceramic pre-layer was loose, the active Si could diffuse through the pre-layer and react with the TZM Mo-based substrate to generate Mo-silicides by Reactions (11)–(13). The sintering temperature was far below the melting point of ZrB_2_, and there was no chemical reaction between ZrB_2_ and Si at the sintering temperature. According to the XRD patterns in Figure 1 and Figure 5a, the ZrB_2_ particles were restrained from the ceramic pre-layer to the as-packed coating. Note that a MoB diffusion layer (layer III in Figure 7a) was formed between the MoSi_2_ layer (layer II in Figure 7a) and the TZM substrate. During siliconization, the consumption of MoB in the ceramic pre-layer to MoSi_2_ by Reaction (10) released B. The diffusion of B in the coating is much faster than that of Si; thus, the excess B would diffuse toward the coating–substrate interface and reacted with Mo at the interface to form a thin MoB diffusion layer between the coating and substrate by Reaction (14). According to the above analysis, the schematic diagram of the coating preparation is shown in Figure 9.

### 3.2. Oxidation Behavior of the Coating

Figure 10a shows the isothermal oxidation kinetic curves of the coating at 1600 °C. The composite coating performed excellent oxidation resistance and could offer effective anti-oxidation protection for the TZM alloy over 20 h at 1600 °C. The weight gain of the oxidized coating was plotted as a function of oxidation time. As seen in Figure 10a, the coating exhibited continuous weight gain with increasing oxidation time. The mass gain of the sample increased rapidly and approximately linear until the value reached 2.16 mg/cm^2^ after 2 h of oxidation. The overall weight gain of the coating was 5.24 mg/cm^2^ after 20 h of oxidation. The approximate linear relationship between the oxidation time and the square of the mass gain in Figure 10b implied that the mass gain of the coating obeyed a parabolic law. The parabolic rate constant could be calculated by Equation (15) [51]:(15)$${\left(\frac{△m}{A}\right)}^{2}=K_{P}t$$
where △m is the mass change of the coated sample (mg), A is the superficial area (cm^2^) and t is the oxidation time (h), and $K_{P}$ is the oxidation rate constant (mg^2^·cm^−4^·h^−1^). The oxidation parabolic constant $K_{P}$, which corresponds to the slope of the curve, was approximately 1.2811 mg^2^·cm^−4^·h^−1^.

It is reported in previous works [18,19] that the weight gain of the coating is likely to relate to the formation of an oxide scale on the coating surface during oxidation. Chemical reactions of the coating with oxygen were conducted to form an SiO_2_-based oxides scale (containing e.g., SiO_2_, ZrO_2_, B_2_O_3_) when exposed to high-temperature oxidizing environments. However, in the initial stage, the oxide scale formed on the coating surface was incomplete and discontinuous, which could not completely cover the substrate, and the coating phases were still exposed to air and could react with oxygen. Consequently, rapid oxidation was observed in the initial stage. When a compact protective oxide scale was formed on the surface of the coating, the oxidation rate was under the control of the permeation of oxygen through the oxide scale; subsequently, the mass gain of the coatings slowed down.

### 3.3. Phase Composition of the Oxidized Coating

Figure 11a shows the XRD pattern of the surface of the MoSi_2_-ZrB_2_ coating after 2 h of oxidation. The characteristic diffraction peaks of SiO_2_ (JCPDS No. 70-2538, 70-2539), ZrO_2_ (JCPDS No. 79-1796), and Mo_5_Si_3_ (JCPDS NO.65-2783) were detected. There was a broad hump in the pattern in Figure 11a, which could be assigned to amorphous SiO_2_. The XRD result in Figure 11a indicates the oxidation of MoSi_2_ and ZrB_2_ to form Mo_5_Si_3_, SiO_2_, and ZrO_2_. The XRD pattern of the surface of the single MoSi_2_ coating after 2 h of oxidation is given in Figure 11b. The result indicates that the phases of the MoSi_2_ coating after 2 h of oxidation was comprised of Mo_5_Si_3_ (JCPDS NO.65-2783), MoSi_2_ (JCPDS NO.80-0544) and SiO_2_ (JCPDS NO.70-2538). There was also a broad hump in the pattern of single MoSi_2_ after 2 h of oxidation, which was amorphous SiO_2_ as well. Since the oxide scale formed on the surface of the MoSi_2_ coating after oxidation at 1600 °C for 2 h was relatively thin, the MoSi_2_ phase in the residual coating beneath the scale was also detected.

### 3.4. Microstructure Evolution of the Coating during Oxidation

#### 3.4.1. Surface Morphology

Figure 12a,b reveal the surface morphology of the single MoSi_2_ coating and MoSi_2_-ZrB_2_ composite coating after oxidation at 1600 °C for 2 h, respectively. Both coatings after oxidation exhibited a compact and smooth surface compared to that of the as-prepared coating. A glassy oxide scale embedded with grayish-white particles (Spot 1) was formed on the surface of the MoSi_2_-ZrB_2_ composite coating after oxidation, as shown in Figure 12b. However, a compact glassy oxide scale without dopants was observed on the surface of MoSi_2_, as shown in Figure 12a. Figure 12c–f display the element mapping of the surface of the MoSi_2_-ZrB_2_ coating after 2 h of oxidation. It can be seen that the glassy phases were rich in Si and O (SiO_2_-based glassy oxide), while the grayish-white particles were rich in Zr and O. The Zr-rich regions in Figure 12d match well with the grayish-white particles regions in Figure 12b. WDS analysis combined with XRD results in Figure 11 confirmed that the grayish-white particle was ZrO_2_ and the glassy oxide phase was SiO_2_. It has been widely reported that the existence of ZrO_2_ on the SiO_2_-based oxide scale was beneficial to improve the anti-oxidation property of the oxide scale, attributing to its excellent thermal stability, low oxygen permeability, and the enhancement of thermal mismatch between the oxide and coating layers [20]. B was not detected in the oxide scale by WDS, mainly due to volatilization at 1600 °C (above boiling point of B_2_O_3_).

#### 3.4.2. Cross-Sectional Microstructure

Figure 13a,b show the cross-sectional microstructure of the MoSi_2_-ZrB_2_ coating and MoSi_2_ coating after 2 h of oxidation at 1600 °C. It can be seen that both coatings exhibited a multi-layer structure. It can be seen in Figure 13a that a five-layer structure (marked as layer I, II, III, IV, V, respectively, from outside to inside) was developed in the cross section of the MoSi_2_-ZrB_2_ coating after oxidation. Combined with XRD and WDS analysis, the five-layer structure was identified to be composed of a dense SiO_2_-based oxide scale with dispersed ZrO_2_ white particles (layer I), a Mo_5_Si_3_ diffusion layer between the coating and the oxide scale (layer II), a MoSi_2_ layer (layer III), a Mo_5_Si_3_ diffusion layer beneath the MoSi_2_ layer (layer IV) and a MoB layer between the coating and substrate (layer V), with a thickness of about 11.0 μm, 8.22 μm, 28.8 μm, 18.5 μm, and 11.5 μm, respectively. According to Figure 13a and the element mappling in Figure 14c,d, some ZrO_2_ white particles distributed on the top of the oxide scale, which is in agreement with the surface observation in Figure 11b. Aggregation tends to take place among the ZrO_2_ particles. The formation of the outer Mo_5_Si_3_ layer (layer II in Figure 13a) was attributed to the oxidation of MoSi_2_, whereas the formation of the inner Mo_5_Si_3_ layer (layer IV in Figure 13a) was derived from inward diffusion of Si from the coating to the substate [18].

Cracks were found in the MoSi_2_ layer and Mo_5_Si_3_ layer in the oxidized MoSi_2_-ZrB_2_ coating. The formation of the cracks was attributed to thermal stresses induced by the CTE mismatch at the coating–substrate or coating–oxide scale interface. However, no obvious oxygen diffusion into the substrate through the cracks was observed, which indicated that the cracks were sealed by the glassy oxides. The cracks developed into through-thickness cracks perpendicular to the substrate in the MoSi_2_ layer (layer III in Figure 13a) and stopped growing or transformed into transverse cracks in the Mo_5_Si_3_ layer (layer IV in Figure 13a), as shown in Figure 12a. A similar phenomenon was observed for the single-MoSi_2_ coating, which indicates that the Mo_5_Si_3_ diffusion layer beneath the MoSi_2_ layer played an important role in hindering the crack propagation toward the substrate for both coatings.

Similarly, as shown in Figure 13b, a four-layer structure (marked as layer I’, II’, III’, IV’, respectively, from outside to inside) was observed in the oxidized MoSi_2_ coating: a thin SiO_2_ oxide scale (layer I’), a Mo_5_Si_3_ diffusion layer between the coating and oxide scale (layer II’), a MoSi_2_ layer (layer III’), and a Mo_5_Si_3_ diffusion layer beneath MoSi_2_ layer (layer IV’), with a thickness of 8.2 μm, 10.0 μm, 15.8 μm, and 24.8 μm, respectively.

It is noteworthy to contrastively analyze the thicknesses of the Mo_5_Si_3_ diffusion layers between the coating and the substrate (layer IV in Figure 13a about 18.5 μm in thickness, layer IV’ in Figure 13b about 24.8 μm in thickness) and the residual MoSi_2_ layers (layer III in Figure 13a about 28.8 μm in thickness, layer III’ in Figure 13b about 15.8 μm in thickness). It is clear that, after 2 h of oxidation, the thickness of the Mo_5_Si_3_ diffusion layer between the coating and the substrate for the MoSi_2_-ZrB_2_ coating was larger than that for the MoSi_2_ coating while the thickness of the residual MoSi_2_ layer for the MoSi_2_-ZrB_2_ coating was smaller than that for the MoSi_2_ coating. It can be deduced that the MoB layer played an important role in preventing the diffusion of Si from the coating to the substrate.

Figure 15 shows the surface and the cross-sectional microstructure of the MoSi_2_-ZrB_2_ coating after oxidation for a longer time (10 h) at 1600 °C. It can be seen that the cross-sectional microstructure of the composite coating after 10 h of oxidation was composed of five layers, which was similar to that after 2 h oxidation. Figure 16 shows the XRD pattern of the composite coating after 10 h of oxidation. The marks show the positions of the peaks corresponding to Mo_5_Si_3_, SiO_2_, ZrO_2_, ZrSiO_4_, and Al_2_O_3_. The formation of ZrSiO_4_ demonstrated the chemical reaction between the formed SiO_2_ and ZrO_2_. Moreover, combined with WDS, we concluded that the phase composition of each layer in the cross section of 10 h was also the same with that of 2 h, although the thickness differed. The thickness of SiO_2_-based oxide scale (layer I in Figure 15) reduced to ~6.8 μm, while the Mo_5_Si_3_ diffusion layer (layer IV in Figure 15) thickened to ~113.6 μm. Compared to the MoSi_2_-based coating without the MoB diffusion barrier in our previous works [18,19], it is worth mentioning in this work that even after 10 h of oxidation, the MoSi_2_ layer was not completely converted to Mo_5_Si_3_, which could still provide the reservoir of Si for the formation of the compact SiO_2_-based glass. It is inferred that the MoB diffusion barrier played an important role in restraining the depletion of MoSi_2_ by hindering the Si diffusion toward the substrate, which proved to be beneficial to the long life of the coating. It also can be seen that the composite coating was retained intact and the SiO_2_-based glassy oxide scale was still protective since no through-thickness crack was formed in the oxide scale, or the coating and the oxide scale was well adhered to the coating.

### 3.5. Antioxidation Mechanism of the Composite Coating

Figure 17 displays the schematic oxidation mechanism of the MoSi_2_-ZrB_2_ coating and single MoSi_2_ coating at 1600 °C in air. Reactions (16)–(20) are possible to conduct when the MoSi_2_-ZrB_2_ coating samples were exposed to an oxidizing environment at 1600 °C. As mentioned in previous work [18,19], the standard Gibbs free energy (based on one mol oxygen) of Reaction (16) is more negative than that of Reaction (17). Thus, the oxidation of MoSi_2_ to form Mo_5_Si_3_ and SiO_2_ by Reaction (16) was dominant. As a result, a SiO_2_-based oxide scale was formed on the coating surface; simultaneously, a thin Mo_5_Si_3_ layer was generated between the MoSi_2_-ZrB_2_ layer and the oxide scale due to the selective oxidation of Si. Meanwhile, the oxidation of ZrB_2_ could produce ZrO_2_ and B_2_O_3_ by Reaction (18). Since B_2_O_3_ would volatilize rapidly at the oxidation test temperature (1600 °C), the oxide scale was composed of ZrO_2_ and SiO_2_. With the increase in the oxidation time, ZrO_2_ and SiO_2_ would react to generate ZrSiO_4_ by Reaction (19), and a compact glassy SiO_2_-ZrO_2_-ZrSiO_4_ oxide scale would form and completely cover the coating surface. The pores and pits in the coating could be filled; SiO_2_, ZrO_2_, and ZrSiO_4_ all have low oxygen diffusion coefficients and the inward diffusion of oxygen could be effectively suppressed [52]. It has been reported that the formation of a composite SiO_2_-based oxide scale embedded with the thermal stable ZrO_2_ and ZrSiO_4_ is believed to enhance the high-temperature oxidation resistance of the MoSi_2_-based coating. SiO_2_-ZrO_2_-ZrSiO_4_ could develop a special glass-ceramic skeleton structure on the surface of the coating and possessed the peculiarities of the stress tolerance of a glass scale and the structural stability of the ceramic phase, in which the ZrO_2_ and ZrSiO_4_ particles have a pinning effect on the SiO_2_ glassy oxide. Mo_5_Si_3_ would be generated beneath the MoSi_2_ layer, which was derived from the inward diffusion of Si (MoSi_2_) from the coating to the substrate according to Reaction (20). At the same time, the MoB layer formed in the as-packed coating could retard the degradation of MoSi_2_ into Mo_5_Si_3_ low-Si silicide by hindering the inward diffusion of Si from the coating to the substrate, which is beneficial to prolonging the service life of MoSi_2_-based coating at high temperatures.
5/7MoSi_2_ + O_2_ = 1/7Mo_5_Si_3_ + SiO_2_
(16)
2/7MoSi_2_ + O_2_ = 2/7MoO_3_ + 4/7SiO_2_
(17)
2/5ZrB_2_ + O_2_ = 2/5ZrO_2_ + 2/5B_2_O_3_
(18)
SiO_2_ + ZrO_2_ = ZrSiO_4_
(19)
MoSi_2_ + 7/3Mo = 2/3Mo_5_Si_3_
(20)

## 4. Conclusions

In this work, ZrB_2_ particles were introduced to the MoSi_2_ coating on the TZM alloy by the in situ reaction of Zr and B_4_C to form a novel MoSi_2_-ZrB_2_ composite coating by a two-step process, including slurry sintering of Mo-Zr-B_4_C followed by pack siliconizing. A ceramic pre-layer composed of ZrB_2_ and MoB was obtained on the TZM substrate by the sintering of Mo, Zr, and B_4_C. After pack siliconizing, the coating was converted into a multi-layer structure, consisting of a MoB inner layer and a MoSi_2_-ZrB_2_ outer layer. The oxidation experiments revealed that the composite coating could provide excellent oxidation-resistant protection for the TZM alloy substrate. The oxidation kinetic curve of the composite coating followed the parabolic rule, and the weight gain of the coated sample after 20 h of oxidation at 1600 °C was only 5.24 mg/cm^2^. When exposed to high-temperature oxidizing environments, the coating rapidly generated a dense and continuous SiO_2_-ZrO_2_-ZrSiO_4_ oxide scale with high thermal stability and low oxygen permeability, which hindered the inward diffusion of oxygen at high temperatures. Concurrently, the MoB inner diffusion layer acted as a diffusion barrier and could hinder the diffusion of Si inward with regard to the TZM alloy substrate and improve the long life of the coating. The excellent oxidation resistance of the coating was mainly attributed to the formation of a compact SiO_2_-ZrO_2_-ZrSiO_4_ composite oxide scale with low oxygen permeability and high thermally stability on the coating surface and the diffusion barrier effect of the MoB layer, which hindered the degradation of MoSi_2_.

## Figures and Tables

**Figure 1 materials-16-05634-f001:**
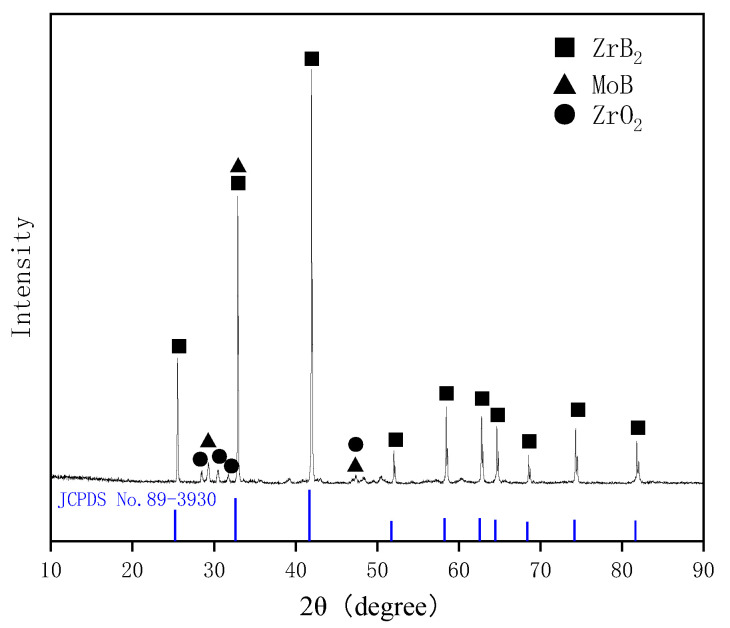
XRD pattern of the surface of the pre-fabricated ceramic layer.

**Figure 2 materials-16-05634-f002:**
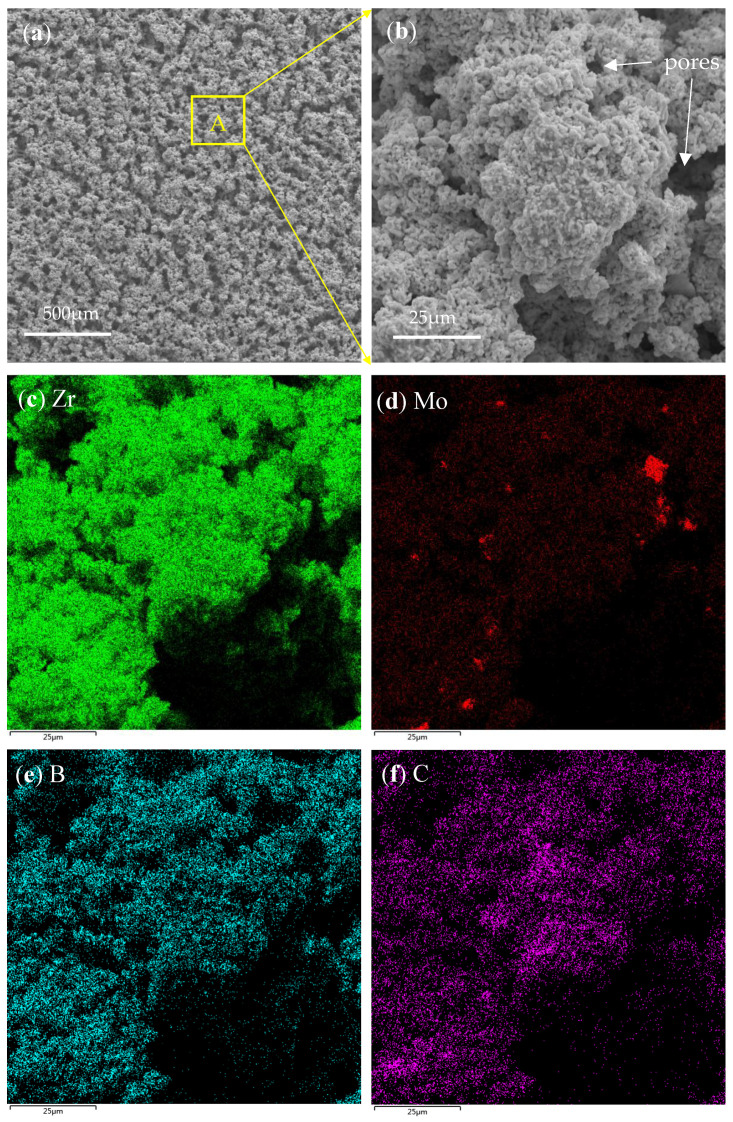
(**a**) surface morphology, (**b**) enlarged view of area A in (**a**), (**c**–**f**) element mapping of the surface of the ceramic pre-layer in (**b**).

**Figure 3 materials-16-05634-f003:**
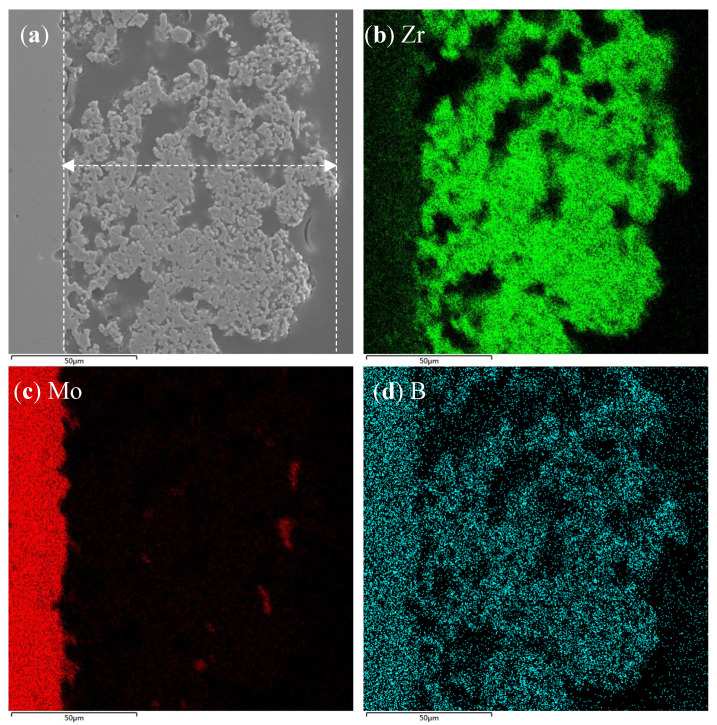
(**a**) microstructure and (**b**–**d**) element mapping of the cross-sectional microstructure of the ceramic pre-layer.

**Figure 4 materials-16-05634-f004:**
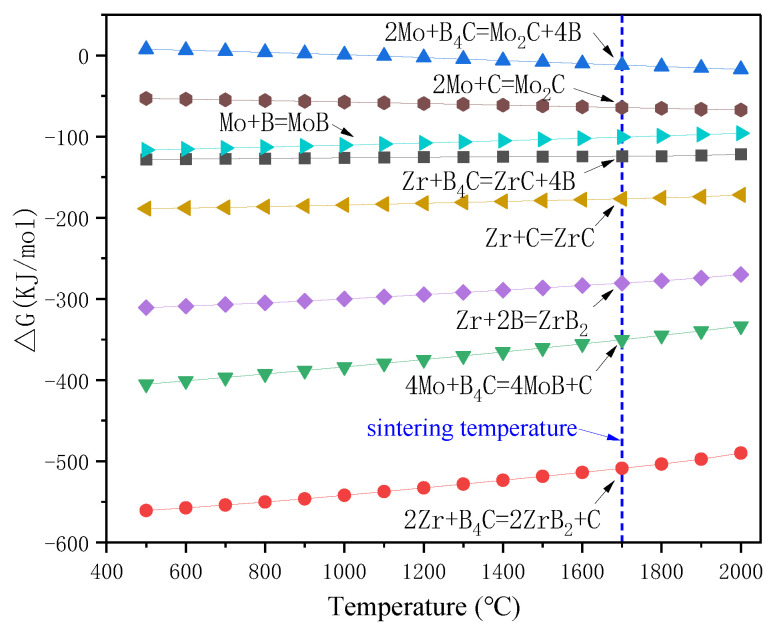
Temperature dependence of the Gibbs free energy of Reactions (1)–(8) from 500–2000 °C.

**Figure 5 materials-16-05634-f005:**
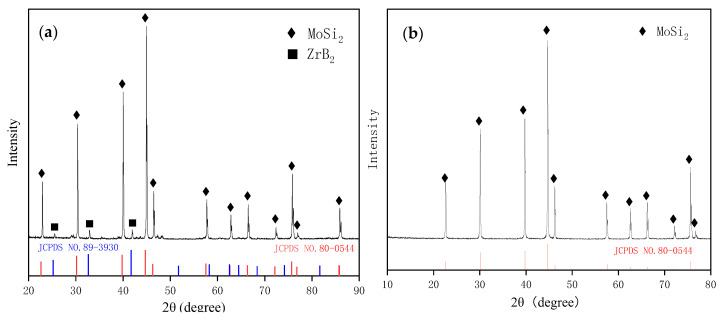
XRD pattern of the surface of (**a**) the composite coating and (**b**) MoSi_2_ coating.

**Figure 6 materials-16-05634-f006:**
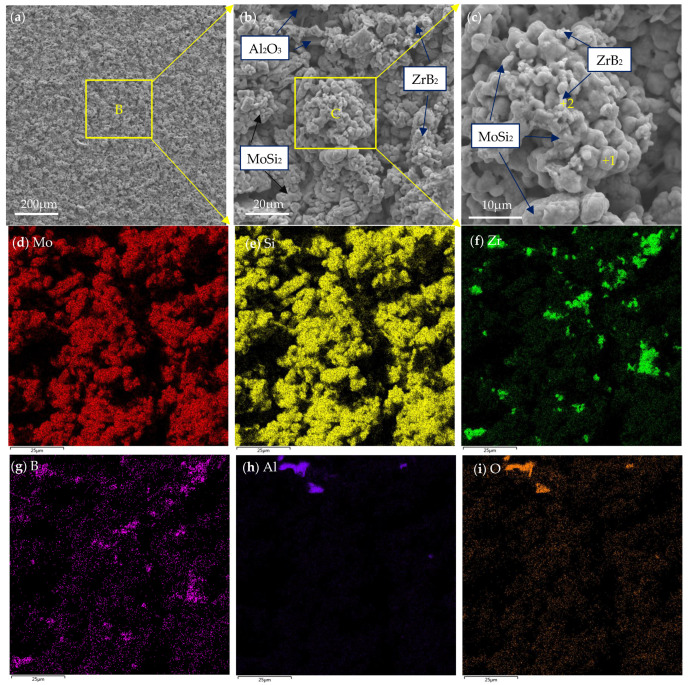
(**a**–**c**) Surface morphology, (**d**–**i**) element mapping of the surface of the MoSi_2_-ZrB_2_ coating.

**Figure 7 materials-16-05634-f007:**
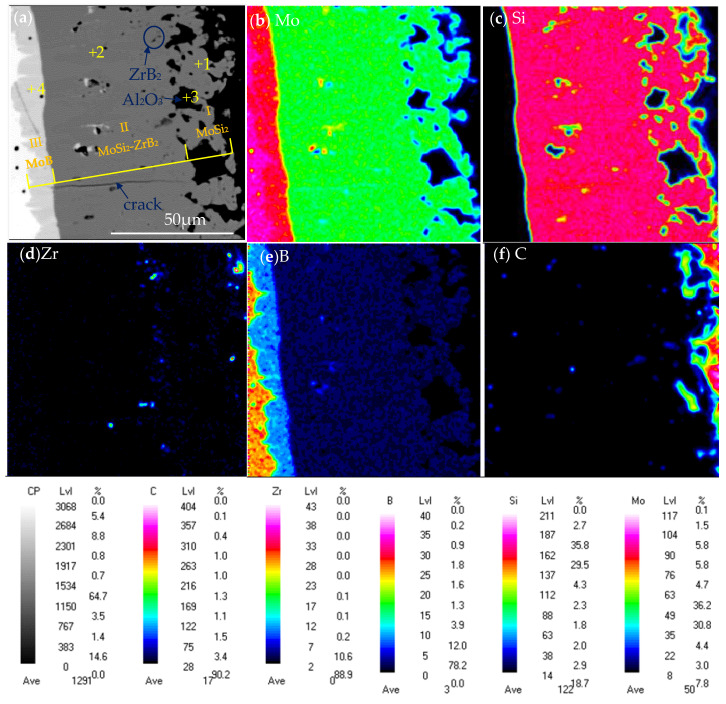
(**a**) cross-sectional microstructure and (**b**–**f**) element mapping of the as-packed composite coating.

**Figure 8 materials-16-05634-f008:**
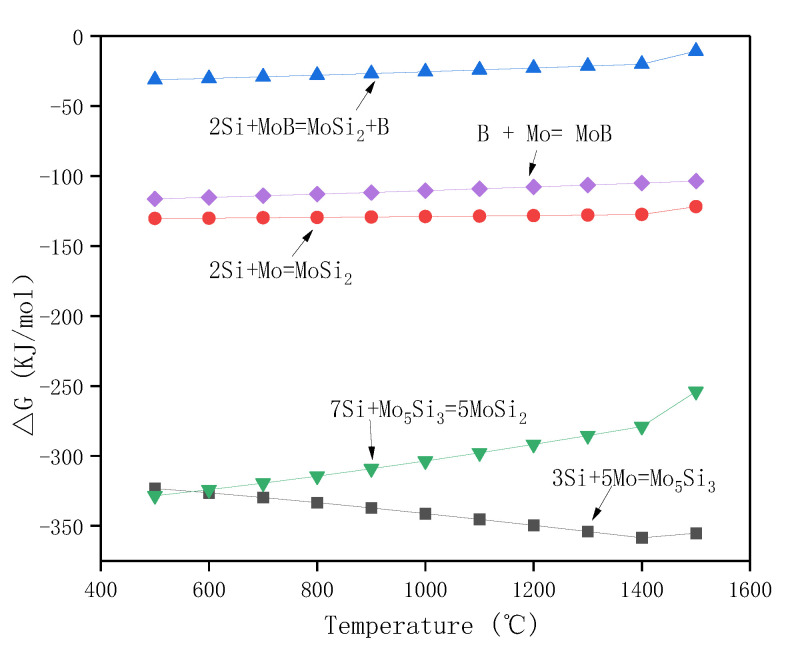
Temperature dependence of the Gibbs free energy of Reactions (9)–(12) from 500–1500 °C.

**Figure 9 materials-16-05634-f009:**
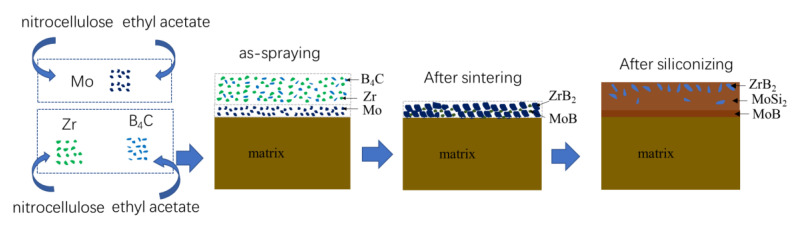
Schematic diagram of coating preparation.

**Figure 10 materials-16-05634-f010:**
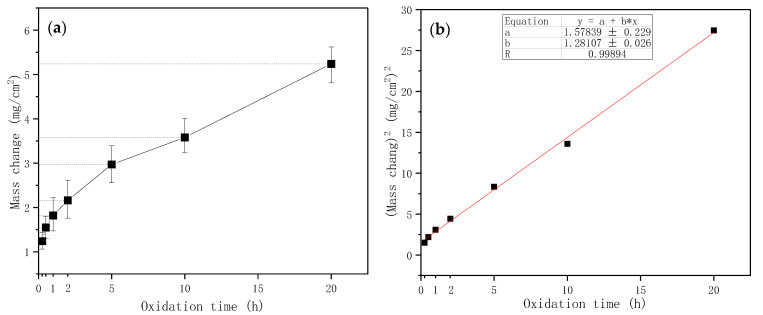
(**a**) The weight gain and (**b**) the square of the wight change of the MoSi_2_-ZrB_2_ coating as a function of duration time.

**Figure 11 materials-16-05634-f011:**
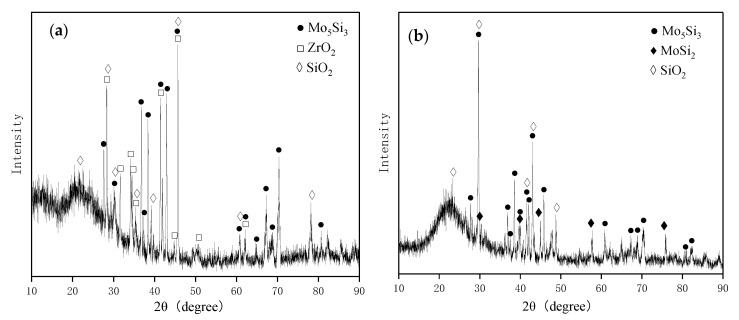
XRD pattern of the surface of (**a**) MoSi_2_-ZrB_2_ and (**b**) MoSi_2_ coating after 2 h of oxidation at 1600 °C.

**Figure 12 materials-16-05634-f012:**
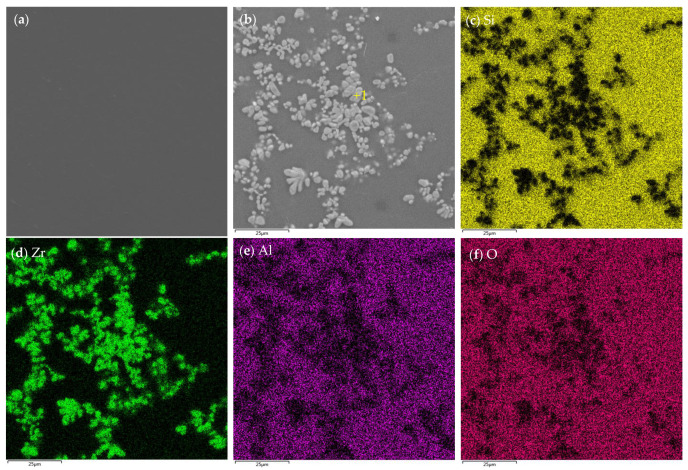
Surface morphology of (**a**) MoSi_2_ coating and (**b**) MoSi_2_-ZrB_2_ coating after 2 h of oxidation and (**c**–**f**) element mapping of the surface of the MoSi_2_-ZrB_2_ coating after 2 h of oxidation in Figure 9b.

**Figure 13 materials-16-05634-f013:**
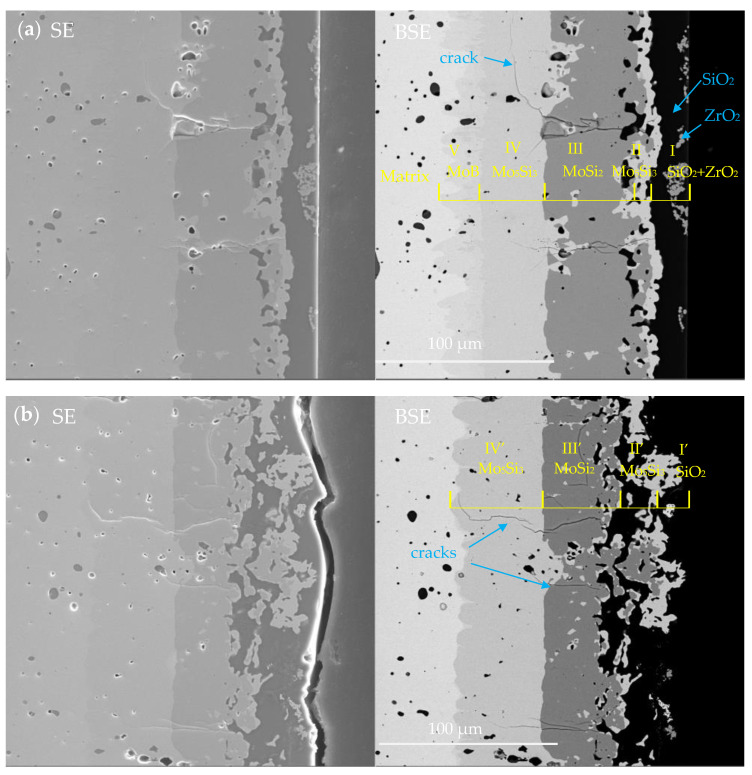
Cross-sectional microstructure of the coating after 2 h of oxidation at 1600 °C (**a**) MoSi_2_-ZrB_2_ coating (**b**) MoSi_2_ coating.

**Figure 14 materials-16-05634-f014:**
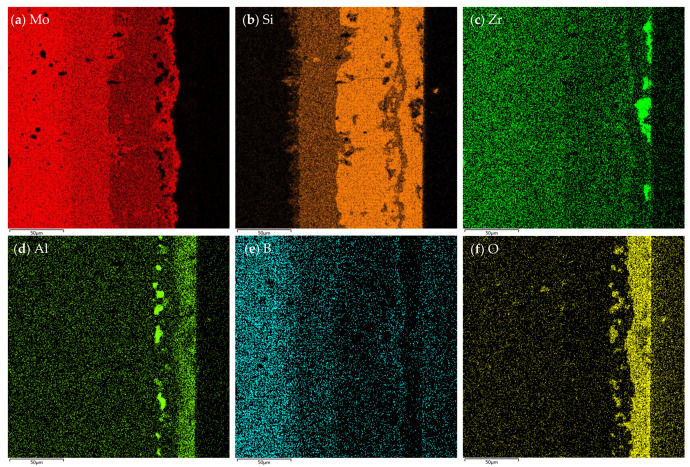
Element mapping of the cross section of the oxidized MoSi_2_-ZrB_2_ coating in Figure 12a. (**a**) Mo; (**b**) Si; (**c**) Zr; (**d**) Al; (**e**) B; (**f**) O.

**Figure 15 materials-16-05634-f015:**
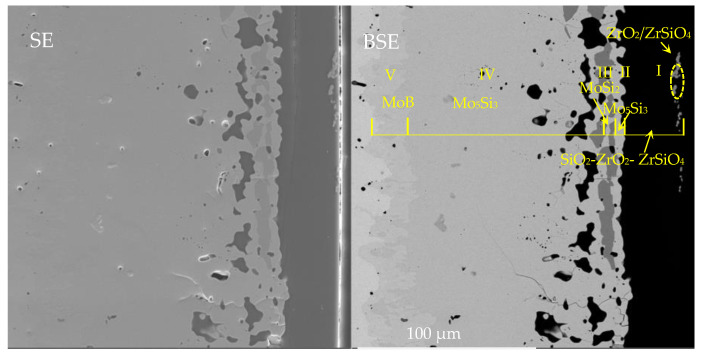
Cross-sectional microstructure of the MoSi_2_-ZrB_2_ coating after 10 h oxidation at 1600 °C.

**Figure 16 materials-16-05634-f016:**
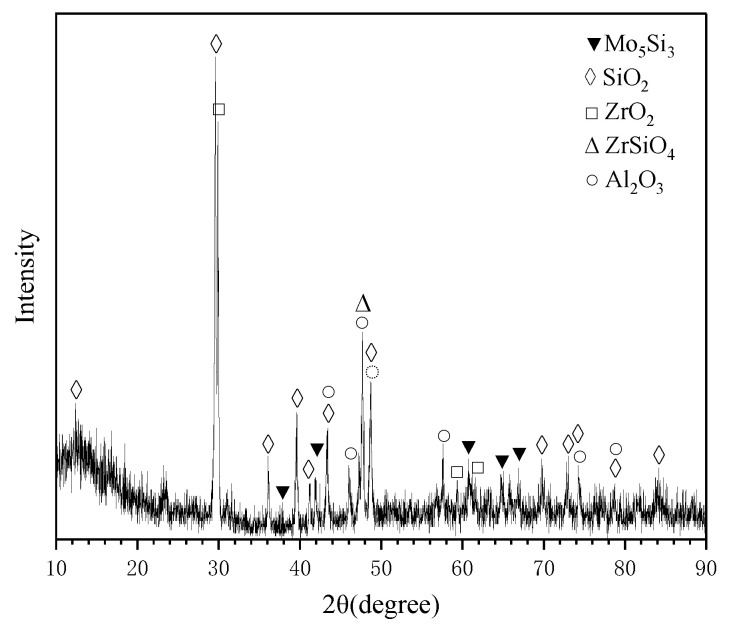
XRD of the surface of the MoSi_2_-ZrB_2_ coating after 10 h oxidation.

**Figure 17 materials-16-05634-f017:**
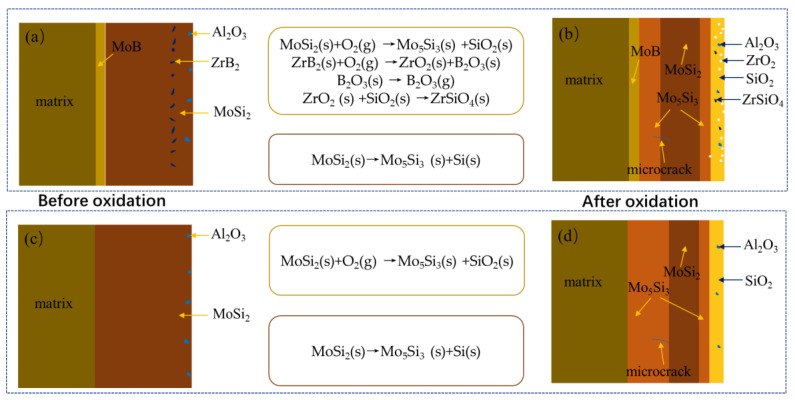
Schematic oxidation mechanism of (**a**,**b**) MoSi_2_-ZrB_2_ and (**c**,**d**) MoSi_2_ coating at 1600 °C in air.

## Data Availability

The data presented in this study are available on reasonable request from the corresponding author.

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
