# Peer review of "Fabrication and Oxidation Resistance of a Novel MoSi2-ZrB2-Based Coating on Mo-Based Alloy"

_materials, 2023, doi:10.3390/ma16165634_

Round 1
Reviewer 1 Report
In this manuscript, a unique MoSi2-ZrB2 composite coating was put on TZM substrate by a two-step process consisting of in-situ reaction of Mo, Zr, and B4C to generate a ZrB2-MoB pre-layer followed by pack siliconizing. The goal was to increase the oxidation resistance of TZM Mo-based alloy. The manuscript is interesting. So, the manuscript may consider for publication after responding to the following comments and revising the manuscript properly.
1. Literature review needs to include several recent, relevant publications (high impact) highlighting their key findings. The current version only discussed general aspects while the review of each from several papers is necessary. You may provide a review summary table consisting of a column for the comments or key conclusions.
2. More recent relevant literature or similar work discussion is mandatory in the introduction section, which is missing in the Introduction. Authors are suggested to add one paragraph in the introduction section by discussing the recent progress and citing similar work.
3. The novelty of the work is missing in the introduction. Authors are suggested to include a separate paragraph discussing the novelty and importance of the present work.
4. Authors are suggested to include a literature review on the recent articles in the introduction section: (i) Effect of sintering conditions on structural and morphological properties of Y- and Co-doped BaZrO3 proton conductors. (ii) Structural, optical, magnetic, and enhanced antibacterial properties of hydrothermally synthesized Sm-incorporating α-MoO3 2D-layered nanoplates. (iii) Visualization of hydrogen isotope distribution in yttrium and cobalt doped barium zirconates.
5. Provide a more appealing title with no acronyms in a precise and concise manner.
6. Omit trivial information.
7. Explain in brief how the present paper differs from the published ones.
8. State-specific objectives.
9. Provide better-quality figures.
10. State the main findings in the conclusions.
Need improvement
Reviewer 2 Report
This study introduces a novel method for generating a MoSi2-ZrB2 composite coating on TZM Mo-based alloy, significantly enhancing its oxidation resistance by forming a stable, low-permeability oxide layer and an internal diffusion barrier. Here are my comments for the manuscript:
1. The sample preparation description would benefit from a flowchart illustrating the specific steps involved in test specimen creation. Additionally, images of the prepared specimen would offer further clarity.
2. The authors need to clarify the reason for choosing 1600 °C as the temperature to evaluate oxidation resistance.
3. How was the sample prepared for characterization? Please introduce the steps involved.
4. In section 3.4.2, the authors should elucidate how they measured the layer thickness. Are the presented values the average thickness observed in the images?
5. Figure 13 requires a correction as it contains two sections labeled as (c).
6. In Figure 14, please denote layer V (MoB).
7. The statement, "combined with WDS, it is concluded that the phase composition of each layer in the cross section of 10 h was also the same with that of 2 h," needs clarification, especially regarding the composition of layer I.
8. Figure 16 appears to have incorrect labeling and needs revision.
9. Mo2B, seen in Figure 15(a) between the substrate and the MoB layer, is not shown in preceding images. The authors should provide an explanation for this.
A thorough proofreading of the manuscript is recommended to fix grammar and spelling errors. For example:
“The coating lifetime of MoSi2 was limited by the Si depletion due to interdiffusion betwwen MoSi2 coating and refractory metals substrate.” Between.
“due to the formation of ZrB2 was priority and rapid” Check grammar.
“the five structure was indensified to be composing of a dense SiO2-based oxide scale with dispersed ZrO2 white particles (layer I)” identified, composed
“However, no obvious oxygen diffusion into the substrate through the cracks” Check grammar.
“It also can be senn that, the composite coating retained intact 468 and the SiO2-baeed glassy oxide scale…” seen.
“It has been reported that 493 the formation of a composite SiO2-based oxide sacle embedded with thermal…” scale
“consisting of a MoB inner layer and an MoSi2-ZrB2 outer layer…” a
Reviewer 3 Report
Comments to the Author(s)
Title: Fabrication and oxidation resistance of a novel MoSi2-ZrB2 based coating on TZM alloy
This manuscript by Zhang et. al., presents the synthesis of MoSi2-ZrB2 composite coating to TZM substrate to provide oxidation resistant. The overall study is interesting. I would like to recommend the article for publication with minor revision. The reference on the oxidation resistance requiring system and properties are not adequate, I would like to suggest to include few more references.
The detailed comments are listed as follows:
1. Define TZM in the beginning?
2. The authors should cite the following references:
https://pubs.acs.org/doi/full/10.1021/acs.jpcc.1c02303
https://doi.org/10.1002/aesr.202100137
3. The author should provide the TGA data to support the stability of the composite.
Round 2
Reviewer 2 Report
Concerns have been addressed in the revised version.